# Review on Surface-Modified Electrodes for the Enhanced Electrochemical Detection of Selective Serotonin Reuptake Inhibitors (SSRIs)

**DOI:** 10.3390/mi14071334

**Published:** 2023-06-29

**Authors:** Simone C. L. Barry, Candice Franke, Takalani Mulaudzi, Keagan Pokpas, Rachel Fanelwa Ajayi

**Affiliations:** 1SensorLab Laboratories, Chemistry Department, University of the Western Cape, Bellville 7535, South Africa; 2Biotechnology Department, Life Sciences Building, University of the Western Cape, Bellville 7535, South Africa

**Keywords:** SSRIs, sensors, electrochemical detection, electrode surface modification, functional material, voltammetry

## Abstract

Selective serotonin re-uptake inhibitors (SSRIs) are one of the most commonly prescribed classes of antidepressants used for the treatment of moderate to severe depressive disorder, personality disorders and various phobias. This class of antidepressants was created with improved margins of safety. However, genetic polymorphism may be responsible for the high variability in patients’ responses to treatment, ranging from failure to delayed therapeutic responses to severe adverse effects of treatment. It is crucial that the appropriate amount of SSRI drugs is administered to ensure the optimum therapeutic efficacy and intervention to minimise severe and toxic effects in patients, which may be the result of accidental and deliberate cases of poisoning. Determining SSRI concentration in human fluids and the environment with high sensitivity, specificity and reproducibility, and at a low cost and real-time monitoring, is imperative. Electrochemical sensors with advanced functional materials have drawn the attention of researchers as a result of these advantages over conventional techniques. This review article aims to present functional materials such as polymers, carbon nanomaterials, metal nanomaterials as well as composites for surface modification of electrodes for sensitive detection and quantification of SSRIs, including fluoxetine, citalopram, paroxetine, fluvoxamine and sertraline. Sensor fabrication, sensor/analyte interactions, design rationale and properties of functional material and the electrocatalytic effect of the modified electrode on SSRI detection are discussed.

## 1. Introduction

Depression is among one of the most prevalent psychiatric disorders and social issues [1] affecting men and women of all ages [2,3]. It is one of the major causes of morbidity and is associated with increased mortality [3]. The World Health Organisation (WHO) estimated that 280 million people globally are affected by the disease. Owing to the COVID-19 pandemic, the rate of depression and anxiety disorders has increased by 25% worldwide [4], thereby increasing the prescription of antidepressants [5]. Antidepressants such as tricyclic antidepressants (TCA), monoamine oxidase inhibitors (MAOIs), serotonin-noradrenaline reuptake inhibitors (SNRIs) and selective serotonin reuptake inhibitors (SSRIs) are available [6]. Antidepressant drugs vary significantly in their effect on individuals [7,8].

Selective serotonin reuptake inhibitors (SSRIs) have become the first choice and among the most prescribed treatments [9] of moderate to severe depressive illness, various phobias, personality disorders and illnesses related to serotonin deficiencies nationally and worldwide [10]. SSRIs act by selectively inhibiting the presynaptic recovery of serotonin or 5-hydroxytryptamine (5-HT) at the serotonin transporter (SERT), causing the increase in serotonin at the postsynaptic membrane in the serotonergic synapse [11,12], thus improving depressive symptoms and the cognitive function of patients [13]. The highly prescribed classes in this category are: fluoxetine (FLX), citalopram (CIT), escitalopram (ESC), paroxetine (PRX), fluvoxamine (FLV) and sertraline (STR) [14]. These SSRIs share a similar mechanism of action. However, they are structurally different with no common motifs which subsequently results in their unique pharmacological differences [12].

These agents can safely and successfully treat depression [15]. However, there is a high variability in response to the drugs where a substantial number of individuals may experience no therapeutic response at all or only a delayed one [16]; those who do respond, may experience severe side effects [16]. Side effects include: nausea dizziness, sweating, tremors [17], agitation, blurred consciousness, gastrointestinal symptoms, insomnia, sexual dysfunction [6], risk of arrhythmias, leukopenia, manic behaviour [18] and suicidal tendencies [13]. Although there are many antidepressants available, the effectiveness of the therapy is still a serious issue [19]. Genetic polymorphisms may have an impact on the response to SSRI treatment where the rate at which drugs are metabolised is associated with variants in the gene [20]. Thus, individuals can be classified as poor, intermediate, extensive, and ultra-rapid metabolisers on the basis of their inherited genetic profiles [16]. The intake of the appropriate dosage of antidepressant is important to ensure that the patient gets the optimum therapeutic effects and that the severe side effects and toxicity are minimised [21]. Therefore, the monitoring of SSRIs is crucial for its clinical applications [22].

A variety of advanced analytical techniques for the detection of SSRIs have been found in the literature which include high performance liquid chromatography (HPLC) with ultra-violet detection, liquid chromatography–mass spectrometry (LC–MS) [23], gas chromatography–mass spectrometry (GC–MS) [24], solid-phase extraction, capillary electrophoresis spectrophotometry, tandem mass spectrometry and nuclear magnetic resonance (NMR) spectrometry [10]. However, these conventional methods are expensive, tedious and often involve cumbersome procedures and pre-treatment steps with limited portability which does not make them reliable for rapid detection and makes them inconvenient for everyday use [25].

As a result of the electroactive properties of SSRIs, electrochemical methods have been used for the detection of these drugs providing adequate insight into the redox properties and metabolism of these compounds [26]. Electrochemical sensors for the detection of SSRIs provide simplicity, high sensitivity and selectivity [27], a broad window of potential [28], stability and reproducibility [29]. These sensors are also low cost and provide a rapid response time with easy miniaturization and construction for portable applications for on-site detection of SSRIs [30].

However, with a conventional bare or unmodified working electrode, the electrochemical detection of SSRIs may incur common problems which include, a low detection of SSRIs in real samples, as a result of low concentration of the analytes, the high over potential at the working electrodes as well as the interference of other biomolecules present in the sample with similar oxidation potential. Considering that SSRIs are absorbed onto the surface of the electrode during the oxidation/reduction reactions, the reproducibility and reuse of the bare electrode is limited.

To overcome these challenges, functional material and/or nanomaterials known for their unique physicochemical and electrical properties have been incorporated into the surfaces of the bare working electrodes providing the modified electrode with new and fascinating properties. These impact strongly on the application of electrochemical sensors and biosensors through better selective and sensitive detection, limited interference of other biomolecules, improved reproducibility and increased usage, and the reduced effects of fouling [31]. Studies in material and nanomaterial sciences have progressed significantly, where materials with controlled dimensions and morphologies as well as surface charges and other physicochemical properties have been fabricated which can then be customised to specific analytes for electrochemical sensing. This review summarises the literature data from 2016 to 2023 that have demonstrated the modification of carbon-based electrodes (glassy carbon electrodes, carbon paste electrodes, screen-printed electrodes, graphite electrodes and carbon fiber electrodes) and metal-based electrodes (pencil lead electrodes, gold electrodes and platinum electrodes) with polymers, carbon nanomaterials, metal materials and nano/composites as functional materials and/or nanomaterials. Very few studies were conducted where only the bare or unmodified working electrode was used for the quantification and detection of SSRIs. These include, boron-doped diamond electrodes, edge-plane graphite electrodes [32], hanging mercury drop electrodes [33] and paper-based electrodes [34].

It is important to fully understand the physicochemical and electronic interactions occurring at the analyte-functional material-based electrode interface in order to explore the overall capabilities of advanced electrochemical sensors [35]. Thus, this review aims to evaluate the use of the functionalised material for the surface modification of the working electrode in electrochemical detection and the quantification of SSRIs in various sample types, as the appropriate functionalised material type has a direct effect on the performance and success of the electroanalysis [26].

## 2. Polymer-Based Modification of Electrode Surface

Polymers are molecules composed of monomers. Conducting polymers are organic substances that possess unique chemical, electrical and physical properties. Their low cost, small dimensions and large surface area with simple preparation steps have enabled researchers to apply them in sensors. The conductivity of conducting polymers was initially observed in polyacetylene, where its instability led to the discovery of polyaniline (PANI), poly(o-toluidine) (PoT), poly(o-anisidine) (PoAN), polypyrrole (PPy), polythiophene (PTh) and polyfluorene (PF). Conducting polymers possess elevated redox properties and electron affinity [36]

### 2.1. Fluoxetine

Molecular-imprinted polymers (MIPs) are considered stable polymers [37] that have been used in electrode surface modification as a promising functional material [38]. As synthetic biomimetic analogues with binding cavities for templates [39], MIPs offer chemical and steric specificity with good sensitivity towards the targeted analyte. They are also economical and easy to produce and are durable. MIP suspensions are commonly synthesised from Nafion and chitosan that allow for binding onto the electrode surface. Chitosan is known for its biocompatible and good adsorption traits [40].

Feroz and team [40] produced a novel photoinitiated MIP for the modification of a glassy carbon electrode (GCE) for the sensitive and selective detection of fluoxetine. Through precipitation polymerisation, reproducible layers of MIP were drop coated onto a GCE and the sensor was applied to blood serum samples. Results confirmed the irreversible nature of fluoxetine with greater current responses and sharper oxidations than a non-imprinted polymer layer. MIP modified GCE showed a selectivity towards fluoxetine in the presence of interfering molecules of similar sizes owing to the affinity of the structural cavities for fluoxetine [40]. The modification of the electrode surface may not only provide a platform for the electrode reaction but can also assist in binding the target analyte.

A PVC membrane with the appropriate sensing element placed on the appropriate transducer is necessary when fabricating an electrochemical potentiometric sensor. However, the PVC membrane requires an intermediate material to improve the effectivity of the ion-to-electron transfer. In this case, conducting polymers with efficient ionic and electrical conductivity have proven to be the best option [41]. Conducting polymers have specific functional groups found on their surfaces that can be oxidised and reduced and, in turn, they can electrocatalyse the oxidation and reduction of the analyte of interest. Through hydrogen bonding, these functional groups have been found to possess a high affinity for the target analyte, ion exchange and electrostatic interaction capabilities as well as reducing fouling by forming a protective layer on the electrode. Because of these properties, they have been widely used in electrochemical sensors [29]. Poly(pyrrole) (PPy) is a quasi-unidimensional polymer where the crosslinking and chains occur during its polymerisation. PPy is an insulator but its oxidized derivatives can be good electrical conductors, where its conductivity ranges from 2 to 10 S/m. Higher conductivities are achievable with larger anions during polymerisation [41].

To quantify fluoxetine in pharmaceutical formulations, Madani [41] introduced a solid state potentiometric sensor coated with conducting polymers. Instead of using a traditional membrane electrode, Madani used a graphite rod as a better ion-to-electron transducer and a sensing layer that can be used separately or as a composite. Pyrrole was electrochemically polymerised onto the graphite rod, forming a layer of poly(pyrrole) (PPy). A thin layer of another polymeric composite composed of poly(vinyl chloride), dibutyl phthalate and ion-pair compound of FLX and phenyl borate was then covered on the treated surface. PPy was incorporated to improve the ion-to-electron transduction capabilities of the electrode. The modified sensor showed superior performance compared to traditional symmetric PVC membrane electrodes or even asymmetric coated wire sensors. An LOD of 0.63 µM, a rapid response time, and a lifetime of two months were the characteristics of the modified sensor. Real sample analyses confirmed the applicability of the sensor in quantification of fluoxetine active ingredients of pharmaceutical formulation with great accuracy and precision. The electrochemical deposition of the conducting layer followed by the physical application of the sensing layer lead to better adherence to the solid contact and improved sensor performance [41].

### 2.2. Citalopram

Izadyar and colleagues [42] demonstrated an inexpensive, novel disposable double-polymer pencil lead electrode (PLE) for the detection of SSRIs (fluoxetine, citalopram and sertraline). As an ion-to-electron transducer, PEDOT-C14 was electrodeposited onto the electrode and coated with a PVC membrane (Figure 1). PEDOT-C14 is capable of being reduced for the transfer of cations at the membrane/water interface. To detect the SSRIs in their cationic forms, Izadyar and colleagues [42] applied ITSV and showed the accumulation of these SSRIs in the PVC membrane from water samples which were exhaustively stripped from the membrane for a highly sensitive detection. The modified electrode exhibited high lipophilicity and an LOD of 35, 45 and 25 nM for fluoxetine, sertraline and citalopram, respectively, in tap and river water [42]. In comparison to previous studies, Izadyar and colleagues showed that the use of ITSV eliminated the need for toxic mercury electrodes and instead used a low cost, modified pencil lead-based sensor with adequate detection limit of SSRIs in tap and river water [42].

For the detection of citalopram, Rebelo et al. [43] demonstrated a sustainable and low cost method for fabricating a voltammetric MIP screen-printed carbon electrode. Through computational studies using molecular dynamics, optimal MIP formulations were predicted with a greener and rational approach, avoiding the use of organic solvents that are accompanied by analytical procedures. The modified sensor revealed a good sensitivity towards citalopram detection as revealed by an LOD of 0.162 µM. The sensor is merited by its imprinting factor of 22 and its excellent specificity towards citalopram [43].

### 2.3. Fluvoxamine

For the detection of fluvoxamine, Soleymanpour and Rezvani [18] fabricated a GCE modified with a PVC membrane and polyaniline (PAN). PAN is a conducting polymer and was used as an electron promoter, transferring the electric charge between the membrane and the conducting substrate, and transduced an ionic signal into an electrical signal causing the phase boundary to be thermodynamically well-defined with a low detection limit of 0.078 µM and without an intermediate internal solution. PAN was easily deposited onto the electrode substrate with a strong adherence to the support, exhibiting good environmental stability. The sensor was also successfully employed in urine, blood and pharmaceutical samples with high precision and accuracy [18].

### 2.4. Sertraline

Cyclodextrin inclusion complexes were employed on gold electrodes to enhance the detection of sertraline. Cyclodextrins were able to enhance the bioavailability of sertraline, by increasing its solubility. Regression equations concluded a sensitivity five times higher in the presence of inclusion complexes of cyclodextrin [44]. Table 1 summarises polymer-based, surface modified electrodes and the analytical techniques used for the detection of various SSRIs, their linear range, and limits of detection, potential interference and application samples.

To summarise, when polymers were used individually or as a single functional material for electrode modification, the polymers did not show great enhancements in the detection of SSRIs when compared to polymer composites, whose synergetic effect resulted in increased detection. The literature has shown that various carbon-based electrodes and metal electrodes have been modified with polymer-based functional material. Most studies have focused on using PVC membrane with conducting polymers. In one study, a graphite rod was used instead of the traditional PVC membrane, due to the detection of drift, potentially as a result of high charge-transfer resistance. However, the electrodes that employed the PVC membrane, showed a greater sensitivity for SSRI detection. The incorporation of cyclodextrins onto gold electrodes also revealed higher sensitivity for SSRI detection. Despite the type of electrode, the functional material plays a vital role in the performance of the sensor. The data shows a variety of conducting polymers and how these polymers influence the performance of the electrode.

## 3. Carbon Nanomaterial-Based Modification of Electrode Surface

Carbon nanomaterials have attracted attention in the field of electrocatalysis for the modification of the surface of bare electrodes, because of their unique physicochemical and biological properties [29]. These nanomaterials come from a variety of sources and possess many morphological shapes [45]. They possess high surface-to-volume ratios and high electrical conductivity [46] with enhanced adsorption, fast electron transfer kinetics, robust mechanical strength [47], extensive chemical stability, biodegradability, biocompatibility and ease of functionalisation in comparison to other nanomaterials [37]. These properties have also enabled carbon nanomaterials as a catalysis probe to exhibit low limits of detection, high sensitivities, a wide linear detection range and sensors capable of being reused [48]. Numerous electroanalytical studies have featured the extraordinary properties of carbon nanomaterials, such as carbon nanotubes, carbon nanoparticles and graphene, that have been used to modify the surfaces of bare electrodes for the electroanalysis of SSRIs [29].

### 3.1. Escitalopram

Well-established methods have been used in the literature for the preparation of carbon-based electrodes and each type comes with advantages and disadvantages [49,50]. The development of solid composite electrodes emerged, consisting of one insulating phase coupled with one conductive phase. These electrodes offer stability, mechanical strength, easy surface renewal and low manufacturing cost [51]. The solid composite electrode, polyurethane–graphene, was first introduced in 2002 [52] and has been used in strategies for biological [53], pharmaceutical [54] and environmental detection [55].

For the detection of escitalopram, Baccarin and team [51] introduced three solid composite electrodes (Figure 2A), a graphite–polyurethane bare electrode (EGPU), the graphene modified electrode (EGPU-GR) and the multi-walled carbon nanotubes (MWCNTs) modified electrode (EGPU-CNTs) [51]. Chronocoulometry studies revealed electroactive areas of 0.065, 0.08 and 0.092 cm^2^ for EGPU, EGPU-GR and EGPU-CNTs, respectively. An irreversible electrochemical behaviour of escitalopram was indicated by CV and, based on the analytical responses between the three electrodes (Figure 2B,C), EGPU-GR was selected for the quantification of escitalopram in subsequent experiments in cerebrospinal fluid and urine due to its greater sensitivity. An LOD of 0.25 µM was achieved by the EGPU-GR sensor including a rapid, selective and reproducible performance [51].

### 3.2. Paroxetine

Gomes and his group [56] introduced a dual electrochemical-sensing device for the multiplexed detection of paroxetine and nonylphenol. The platform comprised two screen-printed electrodes (SPE), either functionalised with green synthesised carbon spherical shells (CSS) or pre-treated with sulfuric acid solution for paroxetine and nonylphenol, respectively. Even though the LOD was not the lowest of reported values, the sensor achieved the multiplexed detection of analytes with high selectivity, as a result of the hydrophilic carbonyl and hydroxyl groups on the CSS surfaces. The high performance of the sensor was also attributed to the combination of sensing layers prepared with the screen-printed technology and the potential range in DPV and SWV techniques. Rapid detection, antifouling properties and interference effects were also achieved by the sensor [56]. Table 2 summarises the carbon nanomaterial surface modified electrode and the analytical techniques used for the detection of SSRIs, its linear range, limit of detection, potential interference and application samples.

## 4. Metal/Metal Oxide Nanomaterial-Based Modification of Electrode Surface

Metal and metal oxide nanoparticles have become a study of great interest due to their catalytic, magnetic, optical and electronic properties. In the fabrication of electrochemical sensors and biosensors, metal nanoparticles have assisted in promoting redox processes [57]. Metals and metal oxides such as nickel, manganese, zirconium, titanium, tungsten, iridium, iron, zinc and copper are suitable matrixes for electrode modification because they have high electrical conductivity, wide electrochemical working window, high biocompatibility, large surface area, low toxicity, chemical and photochemical stability, electrochemical activity, are easier to operate and are capable of online analysis and real-time identification [58,59].

Bimetallic nanoparticles are composed of a primary metal that exhibits high catalytic activity and a secondary metal that further promotes the catalytic activity. Often, bimetallic nanoparticles are capable of producing synergetic catalytic enhancements as a result of their activity, selectivity and stability compared to pristine metal nanoparticles [60].

### 4.1. Escitalopram

An array of nanoparticles has been implemented in the construction of sensors as modifiers. Attia and colleagues [61] constructed a sensor, modifying a carbon paste electrode (CPE) with nickel nanoparticles (Ni) and chloranil (CA). Chloranil is an electron acceptor reagent and was used as an electrocatalytic mediator and an electrode modifier. The incorporation of Ni was reported to increase the active sites, thereby increasing the sensitivity towards escitalopram [61].

### 4.2. Sertraline

Zinc ferrite nanoparticles have gained popularity in the field of nanomedicine due to the lower Zn^2+^ toxicity and the large surface area, fast response and superparamagnetic and high coercivity of iron oxide nanoparticles. Tajik and team [28] reported on the electrocatalytic competency exhibited by zinc ferrite nanoparticles (ZnFe_2_O_4_) and modified screen-printed electrode (SPE) for the enhanced detection of sertraline (Figure 3). For the modified electrode, sertraline oxidation occurred at a potential of about 350 mV less positive than that of an unmodified SPE. An LOD of 0.02 µM was exhibited as well anti-fouling effects. The modified electrode highlighted the improved properties as a result of the incorporation of ZnFe_2_O_4_ nanoparticles with detection of sertraline in real samples [28].

With the extraordinary catalytic properties of metal oxide nanoparticles and low toxicity of lanthanum ions, Mohammadi and group [31] reported a simple, yet economical, electrochemical sensor for the detection of sertraline. The La_2_O_3_/Co_3_O_4_ nanocomposite was prepared by the impregnation of La_2_O_3_ nanoparticles onto the surface of Co_3_O_4_ hexagonal nanosheets and was deposited on an SPE. CV and DPV investigations of sertraline with La_2_O_3_/Co_3_O_4_/SPE indicated significant differences in the electrochemical performance and anodic peak current as that achieved by the unmodified SPE electrode. An LOD of 1 µM was yielded with a linear range of 5.0–400 µM with increasing sertraline concentration. Results showed antifouling properties of the modified electrode towards STR and a decrease in the oxidation product. The prepared La_2_O_3_/Co_3_O_4_/SPE has been successfully used for detecting sertraline in sertraline tablet and urine samples with excellent recoveries [31].

In a similar study, Tajik and Beitollahi [62] combined the properties of lanthanum ions and zinc oxide nano-flowers to modify SPE for the detection of sertraline. The modified electrode exhibited satisfactory catalytic activity in comparison to the unmodified electrode [62]. In comparison to the previous study and the La_2_O_3_/Co_3_O_4_/SPE fabricated by the Mohammadi group [31], the synergetic effect of La^3+^/ZnO nanoflowers exhibited a greater sensitivity towards the oxidation of sertraline. The ZnFe_2_O_4_ modified electrode fabricated by the Tajik group [28] displayed the greatest sensitivity and electrocatalytic performance.

Zaimbashi et al. [63] showed that a graphite-SPE modified with ZnO nanoflowers is capable of producing a sensor with great electrocatalytic activity toward the simultaneous detection of imipramine with sertraline. The incorporation of ZnO nanoflowers led to a significant decrease in the oxidation overpotentials of imipramine and sertraline with large peak separations equivalent to 200 mV as compared to the overpotentials experienced at the bare electrode [63].

In summary, SPE was commonly used and modified with metal-based nanoparticles and nanocomposites for the detection of sertraline. Zinc (Zn) and zinc oxide (ZnO) nanoparticles were the most commonly used nanomaterial used on its own or in a composite for the modification of the SPE. Table 3 summarises metallic nanomaterial surface modified electrodes and the analytical techniques for the detection of various SSRIs, their linear range, limit of detection, potential interferences and application samples.

## 5. Carbon–Metal/Metal Oxide Nanocomposite-Based Modification of Electrode Surface

Nanocomposites are solid materials composed of a combination of nanomaterials which have been extensively applied in medical applications as well as for the detection of drugs [42]. When carbon and metal/metal oxides nanomaterials are combined, the performance of their composite is enhanced owing to the individual properties of each constituent as well as synergistic effects that could be effective in modified electrodes [64]. Their fascinating optical, electrical, physical and chemical properties have led to their emergence. Carbon–metal/metal oxide nanocomposites are designed providing metal/metal oxides with the largest active surface area. Carbon acts as the bone structure, conducting electricity between the electrode and the loaded materials. The metal/metal oxide nanomaterials will then exchange electrons with the analytes. Despite the metal oxide nanomaterial not possessing high electrical conductivity like metal nanomaterials, both nanomaterials possess excellent electrocatalytic ability, resulting in higher sensitivity [45].

### 5.1. Citalopram

Graphene is a two-dimensional carbon nanostructure that has been applied in electrochemical sensor development for its low cost, large surface area, electrical conductivity, high electron mobility and strong mechanical strength and plays an important role in biological sensors. The gold–palladium system is known for its high activity towards electrochemical reactions. The literature has shown the enhanced catalytic oxidation of analytes [65] specifically in drug determination [57].

A gold electrode was modified with graphene and gold–palladium bimetallic nanoparticles (Au–PdNPs) through a simple electrodeposition method for the detection of citalopram. The hybridisation of Au–PdNPs with the graphene platform on the gold electrode surface exhibited excellent electrocatalytic activity towards citalopram oxidation with an oxidation peak current in the linear range of 0.5–50 μM and an LOD of 0.049 μM, with high selectivity and reproducibility. The modified electrode was successfully applied to human plasma for citalopram detection [60].

Multi-walled carbon nanotubes (MWCNTs) have been applied as a functional material on electrodes due to its low-detection limits and show good discrimination against background currents with high sensitivities. When used in combination with metal nanomaterials, the properties and functions of MWCNTs are enhanced as a result of the unique physicochemical properties of metal nanomaterials, where these metal nanomaterials may act as electron catalysts at the surface of the electrode influencing the sensitivity and selectivity [66].

The carbon paste electrode is a blend of graphite powder and a pasting liquid binder that has been used over the past five decades for various sensors and detectors [67]. It has become most popular due to its easy preparation, renewable surface, low cost, easy correction and high sensitivity detection [68]. To improve its features, modifier materials have been used.

To demonstrate this, Ghaedi et al. [67] described the voltammetric determination of trace amounts of citalopram with ZnO nanoparticles and MWCNTs modified carbon-paste electrode (CPE) (ZnO–MWCNT/CPE) using ionic liquid as the binder. ZnO–MWCNT/CPE showed exceptional analytical performance for citalopram detection with an LOD and linear range of 0.005 μM and 0.012 to 1.54 μM, respectively (Figure 4). Compared to reported methods in the literature, the proposed sensor revealed high selectivity with good repeatability and reproducibility [67].

In another study, Keypour et al. [69] functionalised a GCE with Fe_3_O_4_@[(EtO)3Si–L] and MWCNTs. Fe_3_O_4_ nanoparticles were used as an ionophore and in combination with [(EtO)3Si–L]/MWCNTs, the electroactive surface area was enhanced with electron-transfer between citalopram and the electrode, resulting in better sensitivity. A good LOD of 0.0532 µM was noted for citalopram [69].

Metal–organic frameworks (MOFs) are a class of porous polymeric material composed of unique two- or three-dimensional inorganic metal centers with organic linkers. MOFs have been used as suitable electrical substrates for their high surface area, flexibility, high porosity and good thermal stability for electrode modification [70]. Zeolitic imidazolate frameworks (ZIFs) are a subclass of metal–organic frameworks (MOFs) with close similarities to aluminosilicate zeolites. ZIF-8, made up of 2-methyl imidazolate and zinc ions, has become an attractive modifier in the fabrication of electrochemical sensors. To improve its low electrical conductivity, ZIF-8 has been coupled with polymers [71], metal and metal oxide nanoparticles [72] and carbon materials [73]. Reduced graphite oxide (RGO) is a derivative of graphene [74] and has been used In the fabrication of extremely sensitive electrochemical sensors due to its high electronic conductivity, mechanical strength and electron transfer rate [75]. With its layered structural similarities to graphite, graphite carbon nitride (g-C_3_N_4_) is the most resistant carbon nitride substance. It exhibits water resistance, biocompatibility as well as low density due to the van der Waal interaction that occurs between the C-N layers. Its electronic structure and medium band gap give g-C_3_N_4_ a greater chemical and thermal stability than graphite, allowing it to be applied as a photocatalyst and in optical sensors [71].

For the detection of citalopram (CIT) and selegiline (SEL), Karimi-Harandi and colleagues [68] modified a CPE with ZIF-8 dispersed on the surfaces of g-C_3_N_4_ and RGO carbon plates. The individual properties of each nanomaterial and its synergistic effects enhanced the catalytic activity and the surface area at the modified electrode, thereby increasing the electron transfer and, in turn, the sensitivity for CIT and SEL detection. ZIF-8 provided a high surface porosity with RGO and C_3_N_4_ exhibiting good conductivity and water resistance, respectively. The modified sensor exhibited a low LOD of 8 µM and 14 µM for citalopram and selegiline, respectively, with acceptable performance in human samples with the need for pretreatment [68].

### 5.2. Escitalopram

Renewable carbon sources are obtained as by-products from various agricultural materials that have undergone thermochemical conversion [76]. Due to its high surface area, highly functionalised structure and intrinsic electrical conductivity, these materials have been used for electroanalytical applications [77]. Generally, renewable carbons sources undergo pre-treatment steps to ameliorate the adsorption properties and increase the surface area or allow the renewable carbon to act as an appropriate immobilisation platform for biomolecules. These modifications may occur through π–π electron interaction [78], Coulomb interaction [79], hydrophobic interaction, hydrogen bonding [80] and covalent bonding [81]. For improved electronic transport and sensitivity, renewable carbons have been modified with metal nanoparticles [82,83].

To show this, Trindade and team [84] presented a GCE modified with copper nanoparticles (CuNPs) and renewable carbon (RC) from bamboo biomass for the detection of escitalopram, fluoxetine and dopamine. Voltammetric responses for fluoxetine displayed a 1–2-fold increase in the peak current for the RC-CuNP-modified electrode in comparison to the bare GCE and the RC-modified GCE, with escitalopram presenting a 2.5–9-fold increase in peak current. The study achieved a sensor using low cost materials for an environmentally affable and versatile approach in electrochemical analysis [84].

### 5.3. Sertraline

MCM (Mobil composition of matter) is a series of mesoporous materials. From the two porous adsorbents, MCM-48 with its cubic structure has shown to be a more suitable material than MCM-41 in the construction of electrochemical sensors, due to its higher pore volume, surface area, thermal stability, catalytic carrier and better adsorbent in divisions methods [85].

Babaei et al. [85] functionalised a GCE with a composite made of iron oxide nanoparticles, MCM-48 and MWCNTs. The large surface areas of the nanocomposite and negatively charged modified electrode were capable of achieving a very high sensitivity and selectivity for the simultaneous detection of serotonin and sertraline in the presence of interfering uric acid. The sensor also presented a lower electrochemical resistance and bigger redox peak currents than those for the unmodified GCE [85].

Ionic liquids (ILs) comprised of organic and inorganic anions and organic cations have not only been used as a solvent but also as a modifier for electrodes as it displays high ionic conductivity, thermal and chemical stability and negligible vapor pressure [58].

Ehzari and team [58] casted MWCNTs and ionic liquid onto the surface of a GCE, followed by the electrodeposition of p-type semi-conductor, nickel oxide nanoparticles (NiONPs). Two weak anodic peaks were observed for sertraline and clozapine by the GCE, whereas enhanced anodic peaks with an additional two new cathodic peaks were observed by the modified electrode [58]. This study confirmed previous studies that have reported on coupling nickel oxide with carbon materials to improve its electrochemical performance [86,87].

For improved performance, surfactants such as sodium dodecyl sulfate (SDS) have been applied for enhanced electron charge transfer and the accumulation of analytes at the electrode surface [88].

Atty and team [88] combined the catalytic properties of MWCNTs and caesium as well as SDS for the fabrication of a modified CPE for sertraline and paracetamol detection in real samples. The proposed modified electrode displayed a higher anodic peak current of the analytes than the unmodified electrodes [88]. Table 4 summarises carbon–metal/metal oxide nanocomposite surface modified electrodes and the analytical techniques for the detection of various SSRIs, their linear range, limit of detection, potential interferences and application samples.

## 6. Carbon–Polymer Composite-Based Modification of Electrode Surface

To promote the performance of polymers, unique composites coupled with carbon have been prepared. Below are examples of the use of these composites for the detection of various SSRIs.

### 6.1. Fluoxetine

The excellent specificity and sensitivity with high thermal and mechanical stability makes MIP a great prospect for high quality sensing applications for target analytes. However, MIPs do have drawbacks that limit their sensitivity due to a low or the lack of an electrocatalytic effect. To counter these limitations and improve the analytical performance of MIPs, various nanostructured material have been coupled with MIP [37].

Alizadeh and Azizi [89] synthesised MIP nanoparticles for fluoxetine binding through precipitation polymerisation and incorporated these nanoparticles with CPE. A higher response for fluoxetine was obtained from the MIP-CP electrode than from the NIP-modified electrode, indicating that the MIP sites are efficient in recognising the target molecule. For improved sensitivity of the electrode, graphene was incorporated revealing an LOD of 2.8 nM [89].

Furthermore, Ardelean and team [90] showed that an electrode composed of carbon nanofiber-epoxy can be used to detect analytes in water samples simultaneously [90].

Hassan and his team [91] devised a method to integrate MIP beads with the PVC membrane onto SPE with MWCNTs. The sensor revealed enhanced reproducibility, repeatability and stability with application in pharmaceutical formulations of fluoxetine [91].

The Abd-Rabboh group [92] showed that screen-printed potentiometric platforms modified with polymers and MWCNTs are capable of detecting trace amounts of fluoxetine, where MWCNTs provide a high potential stability in the sensors compared to the low potential stability in unmodified sensors. They deemed the electrode polarizable, avoiding the ability to buffer any random tiny charge noise [92].

### 6.2. Citalopram

Voltammetric results revealed that GCE functionalised with cyclodextrin and MWCNTs enhances the oxidation peak current of citalopram, producing an LOD of 44 nM with good recovery and reproducibility [93].

### 6.3. Paroxetine

Reduced graphene oxide and phosphor-tungstate were used to modify a PGE for the trace detection of paroxetine. Oghli and Soleymanpour [94] showed, through differential pulse voltammetry (DPV), that the proposed sensor was highly conductive when compared to the bare PGE, presenting improved electrochemical properties. Owing to the residual oxygen and structural defects of reduced graphene, a higher electrical capacity is promoted, which is brought about by the graphene structure which restores electrical conductivity. Phosphotungstic acid (PWA) facilitates the proton transfer. Paroxetine could be precisely measured up to 0.9 nM by the modified sensor and was permitted to be viable for the determination of paroxetine without pretreatment steps [94].

In a similar study, Oghli and Soleymanpour [95] incorporated MIP to improve the selectivity of the graphene oxide-phosphotungstic modified PGE sensor for the simultaneous detection of paroxetine and sumatriptan (Figure 5). The sol-gel technique was used to immobilize MIP on the electrode surface by forming an inorganic mould. The porous and homogeneous surface created by the sol-gel enhanced the active sites on the electrode. The AdDPV results revealed a better performance with an LOD of 0.7 nM [95] than in their previous study [94]. The proposed sensor was fabricated at a low cost and displayed a reduced analysis time with high efficiency and accuracy of the drugs in real tablet, blood serum and urine samples [95].

### 6.4. Fluvoxamine

For the detection of fluvoxamine in biological and pharmaceutical samples, an SPE was modified with an MIP-integrated fluvoxamine template and PVC. The MIP-modified sensor displayed a greater affinity towards fluvoxamine than the NIP sensor. When compared to the HPLC method, the results were satisfactory, revealing recovery values of 97.4–101.9% [96].

### 6.5. Sertraline

Graphene nanoparticles and MIP were used to modify a Pt electrode for the detection of sertraline in human serum. Through precipitation polymerization, sertraline hydrochloride was used as a template molecule to synthesise MIP with methacrylic acid (MAA) and ethylene glycol dimethyl acrylate (EGDMA), having a high selectivity and sensitivity towards sertraline. A higher adsorption ability was exhibited by the MIP modified sensor than that of the non-imprinted polymer (NIP) sensor. The large surface area and high conductivity brought about by graphene provided improved electrode response and platform of the polymer that makes the graphene-MIP an excellent electrical transducer for direct electrical sensing. An LOD of 7 nM was achieved by the sensor [30].

Khosrokhavar and team [97] fabricated a similar MIP sensor by putting a layer of the MIP/graphene suspension on screen-printed carbon electrodes (SPCEs). They combined the electrical conductivity and high surface area of the graphene nanosheets with the high selectivity of the MIP in conjunction with the properties of the SPEs to detect sertraline. MIP/graphene modified SPCEs displayed a higher adsorption ability than the NIP/graphene modified SPCEs. Under optimal conditions, good sensitivity for sertraline was exhibited with a linear range of 0.005 to 0.075 µM and an LOD of 1.99 nM [97]. Table 5 summarises carbon–polymer composite surface modified electrodes and the analytical techniques for the detection of various SSRIs, their linear range, limit of detection, potential interference and application samples.

## 7. Metal–Polymer Composite-Based Modification of Electrode Surface

Metal and metal oxide nanoparticles dispersed on electrically conducting polymers have gained momentum towards their use in the development of sensors and biosensors. Below are various illustrations of their use for the detection of various SSRIs.

### Fluoxetine

Çorman and team [98] presented a GCE modified with a facile interface imprinting method. Through the copolymerisation of HEMA, MAPA and EGDMA, MIP with fluoxetine templates were developed on the GCE in the presence of zinc oxide (ZnO) immobilized nanoparticles. The etching of ZnO in an acid solution produced a porous structure with recognition sites for fluoxetine on the electrode surface (Figure 6). The incorporation of ZnO was to create the porous structure for an improved electron transfer mechanism and avoid the use of a desorbing agent for the harsh removal of the template. ZnO also provided more active sites, improving the sensitivity of the electrode (Figure 6C). An LOD of 0.00267 nM was achieved [98]. Based on the literature from 2016 to 2023 for SSRI electrochemical detection, this was the only study that used a metal–polymer composite for the modification of the electrode surface. The sensor displayed the best performance in terms of sensitivity, with an LOD in the pico-range. A range which other studies have not achieved based on their functional materials.

Table 6 summarises metal–polymer composite surface modified electrodes and the analytical techniques for the detection of various SSRIs, their linear range, limit of detection, potential interference and application samples.

## 8. Carbon–Metal–Polymer Composite-Based Modification Electrode Surface

### 8.1. Fluoxetine

To measure fluoxetine, Ardelean and team [99] constructed a carbon nanofiber-epoxy electrode (CNFE), functionalised with silver (Ag). Ag and CNFE exhibited a synergic effect improving the sensitivity for FXT detection in comparison to commercial Ag electrodes and CNFE electrodes. Without the detection of possible interfering molecules, the modified electrode exhibited great potential for the detection of fluoxetine in real applications [99].

### 8.2. Citalopram

To achieve the maximum electrocatalytic current of a modified electrode, MOFs have been fabricated. However, most of these materials are electrically insulating, and are unable to provide efficient electron transfer pathways due to their hard metal centers and redox-inactive organic linkers with hard donor atoms. To counter these issues, MOF-based sensors have been coupled and doped with various redox-active compounds (such as metallic species) as well as the hybridisation with highly electron conductive materials such as polymers and carbon nanomaterials and nanocomposites [100]. MOFs have also been developed as proton-conducting materials. Mixed ionic-electronic conductors (MIECs) may also serve in electrochemical sensors as they are materials that are capable of conducting both ions and electrons [101]. MIECs are typically composed of two types of conductors such as PEDOT/PSS and have been used for the sensitive detection of biomolecules.

To show this, Madej et al. [102] fabricated a GCE modified with a nanocomposite composed of a manganese-based metal–organic framework (JUK-2), MWCNTs and gold nanoparticles (AuNPs). JUK-2 MOF showed excellent proton-conductive properties, great stability in aquatic media and was capable of forming a highly homogenous layer on GCE. Because JUK-2 displayed no electrocatalytic properties or possessed electron conductivity, MWCNTs and AuNPs were implemented, providing an enlarged active surface area and the formation of the hybrid nanocomposite which acts as an MIEC. EIS and CV results indicated that the hybrid nanocomposite exhibited MIEC properties. An LOD of 0.011 µM was achieved and the sensor was successfully applied to biological and water samples with recovery rates of 98.6–104.8%. The performance and success of the sensor toward citalopram are attributed to JUK-2 MOF that excellently bound MWCNTs and AuNPs, ensuring their synergistic effects as well as the high reproducibility and accuracy of results. The hybrid nanocomposite possessing both electron and proton conductivity was responsible for the outstanding electrocatalytic activity toward the oxidation of citalopram [102]

For the enhanced detection of citalopram, Aminikhah and team [103] fabricated a GCE modified through in situ electopolymerisation of MIP PPy onto hollow nickel nanosperes (hNiNS) activated with MWCNTs-coupled graphene oxide nanoribbons (AMWCNTs@GONRs). Hollow nanopores or spheres are known to exhibit high surface-to-volume ratios, excellent biocompatibility and electrical conductivity properties due to their morphology. Combining hNiNS with AMWCNTs greatly enhanced the current response to citalopram through an increased rate of electron transfer and enhanced catalytic activity. MIP PPy provided specific binding sites for citalopram, improving the selectivity of the sensor. The sensor achieved an LOD of 0.043 µM [103].

### 8.3. Paroxetine

According to the literature, various amino acids have been used for the production of a layer of conductive polymer with electrocatalytic activity, as well as the formation of additional active sites available for target interaction with analytes, enhancing the sensors’ sensitivity and selectivity. ᴅʟ-Methionine is an amino acid that can easily be electropolymerised onto the surface of an electrode as a porous conducting polymer [104].

For the detection of paroxetine, Al-Mhyawi and team [104] modified GCE with gold nanoparticles (AuNPs) onto which DL-methionine conducting polymeric film was electropolymerised. The high surface area, fast electron transfer ability and high conductivity of AuNPs as well as the high potential of DL-methionine to interact with AuNPs, provided the modified electrode with improved performance, greater sensitivity, and conductivity to detect paroxetine in comparison to the bare electrodes. The sensor was also highly selective towards paroxetine. An LOD of 0.01 nM was exhibited by the modified sensor [104].

### 8.4. Sertraline

The exploration of metallopolymers containing Ni(II)–Ni(III) redox couples have gained great attention owing to their high electrocatalytic activity towards electro-oxidation of various substrates including, amino acids, amines and carbohydrates. Nickel oxyhydride species are formed during the reaction which act as a mediator between the analyte and the electrode surface in electro-oxidation processes. Amino acids such as levodopa (L-3, 4-dihydroxypheny- lalanine, LD) have been used as ligands for the electrochemical deposition of metallopolymers on the electrode surface, specifically in the biomedical fields [64].

For the detection of sertraline, Shoja and team [64] modified a GCE through electropolymerisation of nanostructured Ni(II)–LD (LD: levodopa) complex onto AuNP-bound MWCNTs in alkaline solution. The synergistic effect of the AuNPs and MWCNTs provided an effective large surface area for the immobilisation of Ni(II)-LD nanostructures, which in turn promoted the electrocatalytic oxidation of sertraline as a result of the active Ni(II)/Ni(III) sites. The proposed electrode provided an inexpensive fabrication method with excellent selectivity and sensitivity for sertraline oxidation, a low detection limit with good linear response in wide concentration range and good reproducibility and stability [64].

Habibi et al. [100] fabricated a novel copper-based MOF with S, N co-doped graphene to modify a pencil graphite electrode (PGE). The modified electrode displayed excellent electrocatalytic activity for sertraline oxidation in comparison to the bare electrode as a result of the synergistic effects amongst the copper-based MOF and the graphene, which provide a large specific surface area and enhanced the electron transfer process (Figure 7). The proposed sensor presented an LOD of 0.038 µM, good selectivity and a sensitivity of 0.4557 µA/µM cm^2^ [100]. Table 7 summarises carbon–metal–polymer composite-based surface modified electrodes and the analytical techniques for the detection of various SSRIs, their linear range and limit of detection, potential interference and application samples.

## 9. Bio-Functionalised Polymer-Based Modification of Electrode Surface

### Paroxetine

Ajayi and colleagues [105] fabricated a nanobiosensor composed of nanotubular poly(8-anilino-1-napthalene sulphonic acid) encapsulated with the enzyme, cytochrome P450-2D6, on a gold electrode for the detection of paroxetine. The polymer acted as a solid phase electron mediator due to its π-conjugated backbone. Its ease of synthesis together with its compatibility with enzymes allows polymers to facilitate electron transfer between the redox center of the enzyme and the electrode surface thus producing electronic signals. The biosensor displayed an LOD of 0.002 µM for paroxetine. Paroxetine, in turn, inhibited the activity of P450-2D6, causing a decrease in the signal response for fluvoxamine. This showed that the sensor was capable of creating an immediate response when the analyte comes into contact with the enzyme. The use of fluvoxamine showed a reversible competitive inhibition where the enzyme was only inactivated by paroxetine, as the sensor response would increase in the presence of fluvoxamine. The nanobiosensor displayed excellent catalytic activity for CYP2D6 with long-term use, high sensitivity and stability [105]. Table 8 summarises bio-functionalised polymer-based surface-modified electrodes and the analytical techniques for the detection of various SSRIs, their linear range and limit of detection, potential interference and application samples.

## 10. Concluding Remarks and Future Perspectives

Currently, selective serotonin reuptake inhibitors (SSRIs) are the primary and the most prescribed drugs to treat major depression and depression-related disorders. With the demand for cost-effective, sensitive, accurate and rapid detection of SSRIs, there has been great focus on, and research undertaken towards, developing electrochemical sensors with surface-modified electrodes. This review features a brief overview of recent advances in the electrochemical sensor development for SSRI detection. It has attempted to elucidate the properties and interactions of surface-modified electrodes on target analytes with the significance of the functional material and the potential use of the developed electrochemical sensor. Over the past seven years, significant progress has been made in the development of novel materials and nanomaterials for the modification of electrodes for an efficient, sensitive and selective quantification and detection of SSRIs. According to the published literature, modified electrodes using composites and nanocomposites were more common among sensor development for SSRI detection than individual material because of the synergistic effect of materials for enhanced detection. Research has demonstrated that carbon–metal-based composites as well as carbon–polymer-based composites were the most extensively used functional material for electrode surface modification. In the carbon–metal nanocomposites, MWCNTs were most commonly used in conjunction with the metal and/or metal oxide nanoparticles. MIP was most commonly used in carbon–polymer composites with either graphene or MWCNT material. These materials were shown to provide exceptional detection limits with great selectivity as a result of the synergistic effects that facilitate enhanced catalytic reactions on the target analyte. Metal–polymer-based composites provided the greatest sensitivity for SSRI quantification and detection with the LOD as low as pM. Sertraline was the most detected SSRI using the modified electrodes, which makes sense as sertraline is the most commonly prescribed antidepressant. Most of the studies performed analyses of SSRIs in real samples. The general trend in recent years for detection of SSRIs uses GCE modified mostly with carbon-based functional material and polymers. Metal electrodes have rarely been used as this permits high sensitivity and selectivity with reduced interference, which may be suitable in clinical studies. This also provides high efficiency and a rapid analysis time. The development of new materials and nanomaterials are necessary to enhance surface performance of the electrode in order to achieve ultra-trace analysis of SSRIs in various sample types and, in turn, improve the practicability. In general, the modification of electrode surfaces with functional materials, have great advantages and development prospects in SSRI sensing.

## Figures and Tables

**Figure 1 micromachines-14-01334-f001:**
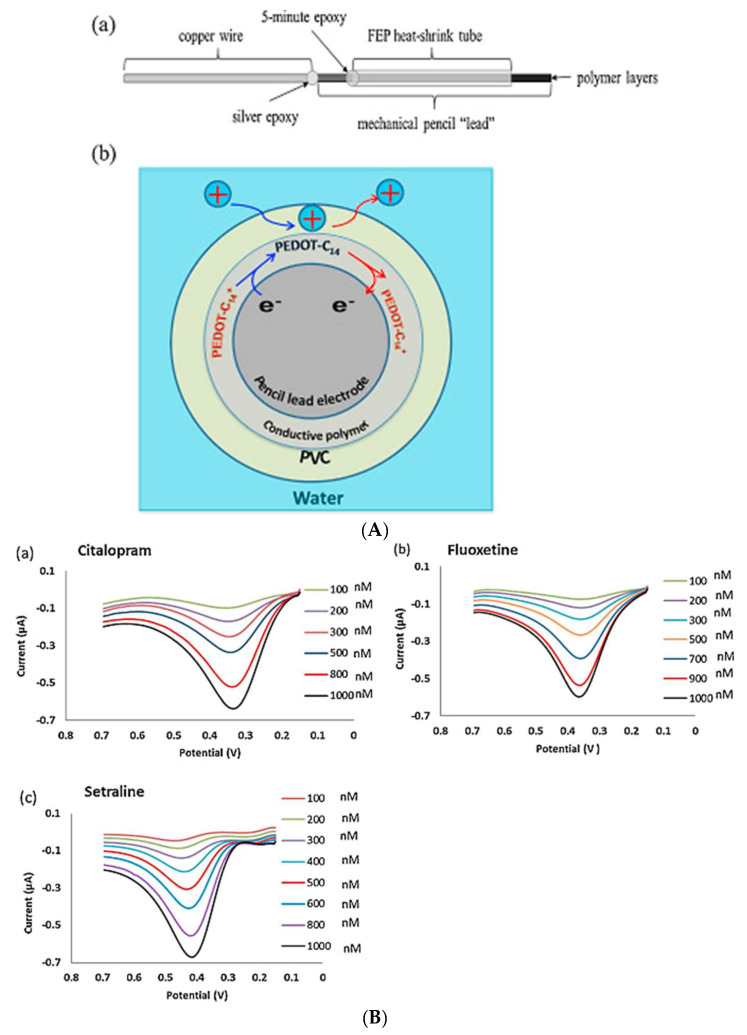
(**A**) (**a**) Fabrication of an unmodified pencil lead electrode, and (**b**) the cross section of the PVC/PEDOT−C14−modified pencil lead electrode for the preconcentration (blue arrows) and stripping (red arrows) of SSRI ions; (**B**) ITSVs of (**a**) citalopram, (**b**) fluoxetine and (**c**) sertraline ions. Reproduced from [42], Copyright 2016, with permission from Elsevier.

**Figure 2 micromachines-14-01334-f002:**
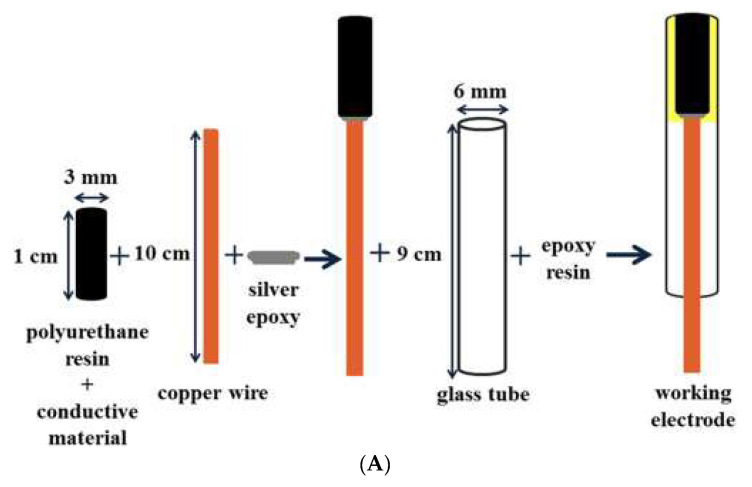
(**A**) Fabrication of the composite electrode. (**B**) CV obtained in the absence and in the presence of EST using the EGPU (**a**), EGPU−GR (**b**) and EGPU−CNTs (**c**) electrodes. Reproduced from [51], Copyright 2018, with permission from Elsevier.

**Figure 3 micromachines-14-01334-f003:**
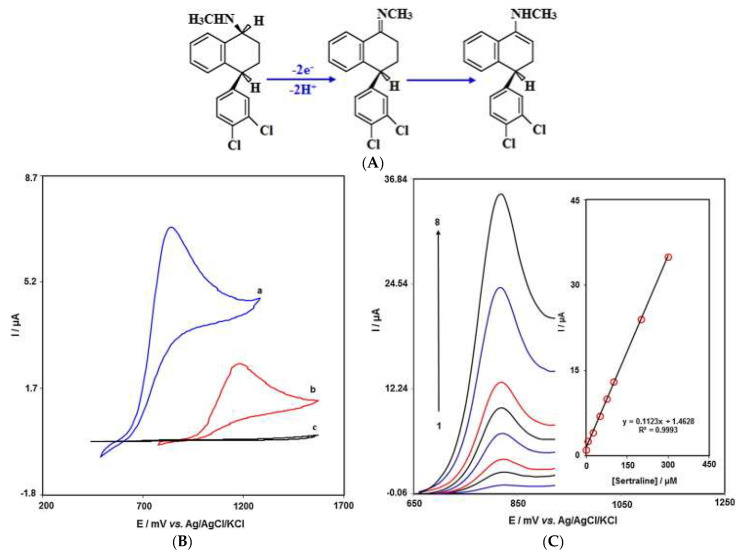
(**A**) Electro-oxidation mechanism of sertraline at ZnFe2O_4_/SPE. (**B**) CVs of (a) ZnFe_2_O_4_/SPE and (b) unmodified SPE in the presence of sertraline and (c) unmodified SPE in the PBS pH 7.0. (**C**) DPVs of ZnFe_2_O_4_/SPE in different concentrations of sertraline. Numbers 1−8 correspond to 0.07, 5.0, 25.0, 50.0, 75.0, 100.0, 200.0 and 300.0 µM of sertraline. The inset shows the plot of the peak current as a function of the sertraline concentration in the range of 0.07−300.0 µM. Reproduced from [28], Copyright 2019, with permission from Elsevier.

**Figure 4 micromachines-14-01334-f004:**
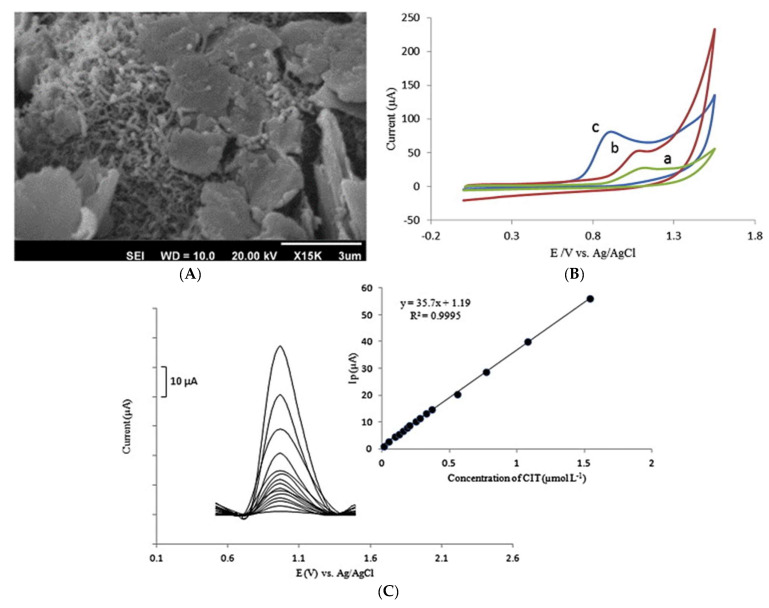
(**A**) SEM image of modified ZnO−MWCNT/CPE electrode, (**B**) CV of CIT obtained in different electrodes (a) CPE_IL_, (b) MWCNT/CPE_IL_ and (c) ZnO–MWCNT/CPE_IL_ surface. (**C**) Net SWV of ZnO−MWCNT/CPE_IL_ solutions with different concentrations of CIT. Reproduced from [67], Copyright 2016, with permission from Elsevier.

**Figure 5 micromachines-14-01334-f005:**
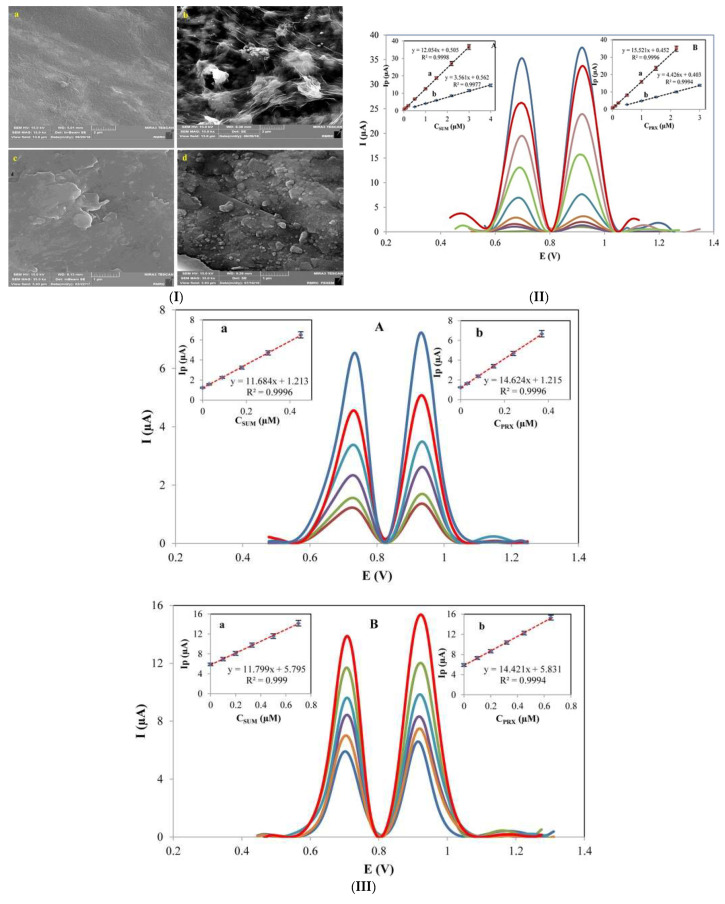
(**I**) SEM image of (**a**) bare PGE, (**b**) PWA/rGO/PGE, (**c**) Sol-Gel/PWA/rGO/PGE and (**d**) MIPP,S/Sol-Gel/PWA/rGO/PGE, (**II**) AdDPV of SUM and PRX in the concentration range of 0.005–3.0 μM at MIPP,S/Sol-Gel/PWA/rGO/PGE. Insets: corresponding calibration curves for (**A**) SUM and (**B**) PRX at (a) MIPP,S/Sol-Gel/PWA/rGO/PGE and (b) NIP/Sol-Gel/PWA/rGO/PGE. (**III**) DPVs of SUM and PRX with various concentrations in (**A**) urine (100 nM of SUM and 80 nM of PRX) and (**B**) blood serum (500 nM of SUM and 400 nM of PRX) samples. Insets: relevant standard addition graphs for (**a**) SUM and (**b**) PRX. Reproduced from [95], Copyright 2016, with permission from Elsevier.

**Figure 6 micromachines-14-01334-f006:**
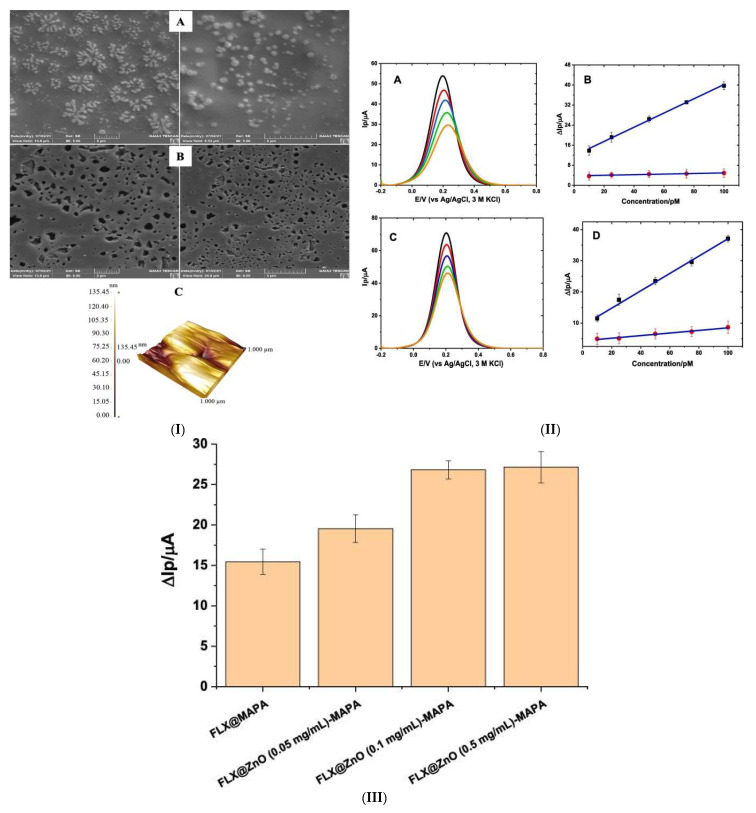
(**I**) SEM images of an FLX imprinted electrochemical sensor (**A**) before and (**B**) after ZnONPs are removed, and (**C**) AFM image with results, (**II**) DPVs of the various concentrations of the FLX: 10 pM (black line), 25 pM (red line), 50 pM (blue line), 75 pM (green line) and 100 pM (orange line) on the FLX@ZnO−MAPA@MIP/GCE in (**A**) standard solution, and (**C**) spiked serum solution. The calibration curve with respect to FLX concentration for MIP (black dots) and NIP (red dots) in (**B**) standard solution, and (**D**) spiked serum solution. (**III**) The effect of ZnO nanoparticles on the MIP-based sensors. Reproduced from [95], Copyright 2023, with permission from Elsevier.

**Figure 7 micromachines-14-01334-f007:**
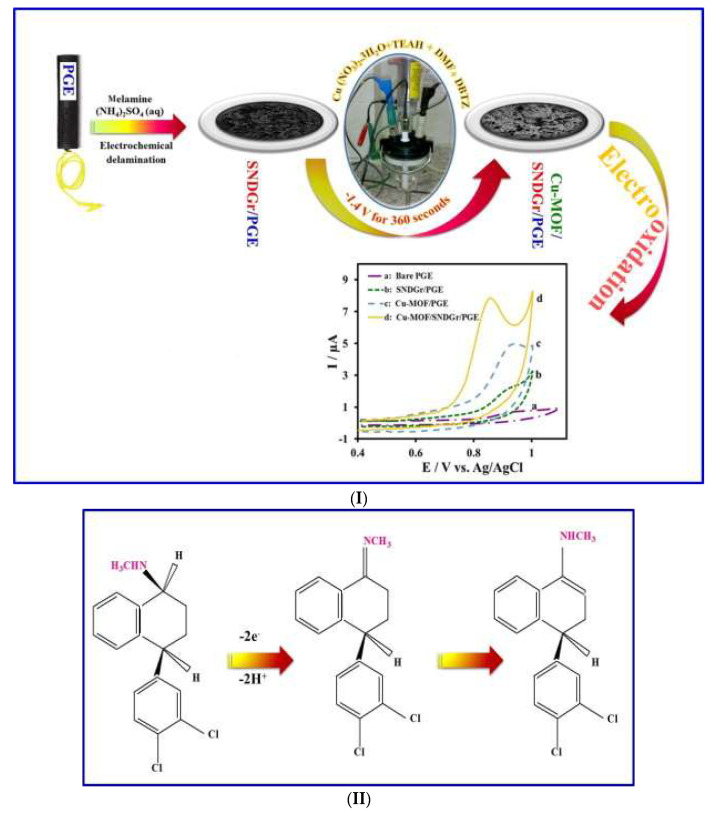
(**I**) Preparation steps of the Cu−MOF/SNDGr/PGE and its application. (**II**) The electrochemical oxidation mechanism of the STR. (**III**) (**A**) DPVs of the Cu−MOF/SNDGr/PGE in 0.05−12.67 µM STR and (**B**) the calibration curve of Ip and STR concentrations in the range of 0.05−12.67 µM. Reproduced from [100], Copyright 2021, with permission from Elsevier.

**Table 1 micromachines-14-01334-t001:** Polymer-based modification of electrode surface.

Analyte	Coating	Technique	Range (µM)	LOD (µM)	Interference	Sample	Ref
FLX	MIP-GCE	CV, SWV, DPV	0.499–33.8	0.33	Biological species	Blood serum	[40]
	Poly(pyrrole)	Potentiometry	1–1000	0.63	Inorganic species and amino acids	FLX.HC:L capsules	[41]
FLX CIT STR	PVC/PEDOT-C14-PLE	ITSV	0.1–1	0.035 0.025 0.045	None	Tap and river water	[42]
FLV	PM-PAN-GCE	Potentiomettry	0.18–1300	0.078	Inorganic species, organic molecules, and amino acids	Human serum and urine	[18]
STR	HPβCD, βCD-Au electrode	CV, SWV	0.1–0.9	0.026	None	Human serum and Sidata tablet	[44]

**Table 2 micromachines-14-01334-t002:** Carbon nanomaterial-based modification of electrode surface.

Analyte	Coating	Technique	Range (µM)	LOD (µM)	Interference	Sample	Ref
ESC	EGPU-GR	DPV, SWV	1.5–12	0.25	Organic molecules and amino acids	Urine and CSF	[51]
PRX	CSS-SPE	SWV	1–100	0.67	Biological species	Tap water	[56]

**Table 3 micromachines-14-01334-t003:** Metallic nanomaterial-based modification of electrode surface.

Analyte	Coating	Technique	Range (µM)	LOD (µM)	Interference	Sample	Ref
ESC	NiCACP	CV, DPV, EIS	1–70	0.2	Amino acids, inorganic cations and sugars	Tablet and urine	[61]
STR	ZnFe_2_O_4_-SPE	DPV	0.07–300	0.02	Amino acids, sugars, inorganic-, organic molecules and biological species	Urine and tablet	[28]
	La_2_O_3_/CO_3_O_4_-SPE	DPV	5–400	1	None	Urine and tablet	[31]
	La^3+/^ZnO-SPE	CV, DPV	0.5–150	0.15	None	Urine and tablet	[62]
	ZnO-NFs/GSPE	DPV	N/A	N/A	Organic and inorganic molecules	Urine and tablet	[63]

**Table 4 micromachines-14-01334-t004:** Carbon–metal/metal oxide nanocomposite-based modification of electrode surface.

Analyte	Coating	Technique	Range (µM)	LOD (µM)	Interference	Sample	Ref
CIT	Au-PdNPs-GR-AuE	CV, SWV, EIS	0.5–50	0.049	Inorganic and organic molecules, amino acids, and sugars	Human plasma and tablets	[60]
	ZnO-MWCNT-CPE	CV, ASWV	0.012–1.54	0.005		Serum, urine and tablets	[67]
	Fe_3_O_4_@[(EtO)3Si-L]/MWCNTs-GCE	CV, DPV	0.3–10,000	0.0532		Blood and tablets	[69]
	ZIF-8/g-C_3_N_4_/RGO-CPE	CV, DPV, EIS	0.009–900	0.008		Blood, urine and tablets	[68]
ESC	GC-RC-CuNP	CV, DPV	0.02–5	0.25	Organic and inorganic molecules, and amino acids	Tap water and urine	[84]
FLX			0.1–10	0.05			
STR	Fe_3_O_4_@MCM-48-SO_3_H/MWCNTs-GCE	CV, DPV	0.05–100	0.015	Inorganic and organic molecules, amino acids, and sugars	Blood and urine plasma	[85]
	MWCNT-IL/NiONPs-GCE	CV, DPAdSV	0.21–85	0.047		Serum and tablets	[58]
	CNT/CsM/SDS-CPE	CV, SWV	0.06–15	0.0092		Plasma and tablets	[88]

**Table 5 micromachines-14-01334-t005:** Carbon–polymer composite-based modification of electrode surface.

Analytes	Coating	Technique	Range (µM)	LOD (µM)	Interference	Sample	Ref
FLX	Nano-MIP/G2-CP	DPV	0.006–0.1	0.0015	Inorganic species, amino acids and sugars	Tablet and plasma	[89]
	CNF	CV, DPV	0–10	0.385	None	Water	[90]
	MIP-MWCNTs-SPE	EIS	0.1–10,000	2.1	Organic and inorganic species, amino acids and sugars	tablet	[91]
	Ionophore I, Ionophore II and Ionophore III-MWCNTs-SPE	Potentiometry	0.2–6.5	5.2, 4.7, 0.2	Ionic and biological species	Blood	[92]
CIT	P(pABSA)/β-CD/MWCNT-GCE	CV, EIS	0.09–100	0.044	Sugars and organic species	Tablet and serum	[93]
PRX	rGo/PWA/PGE	DPV	0.008–6	0.9 nM	Ionic and biological species	Serum, urine and tablet	[94]
	MIPP,S/Sol-Gel/PWA/rGO/PGE	adDPV	0.005–2.2	0.7 nM		Blood, urine and tablet	[95]
FLV	MWCNTs/MIP-SPE	CP, EIS	0.1–10,000	4.8	Inorganic and organic species, amino acids and sugars	Tablet	[96]
STR	Graphene-MIPPtE	CV, DPV	0.01–1	0.007	Inorganic and organic species, amino acids and sugars	Serum	[30]
	MIP/graphene-SPCE	CV, DPV	0.005–0.075	0.002	Dapoxetine, fluoxetine, citalopram, uric acid and ascorbic acid	Serum and tablet	[97]

**Table 6 micromachines-14-01334-t006:** Metal–polymer composite-based modification of electrode surface.

Analyte	Coating	Technique	Range (µM)	LOD (µM)	Interference	Sample	Ref
FLX	FLX@ZnO-MAPA@MIP/GCE	DPV	0.01–0.1 nM	0.00267 nM	Antibiotics	Tap water and serum	[98]

**Table 7 micromachines-14-01334-t007:** Carbon–metal–polymer composite-based modification of electrode surface.

Analytes	Coating	Technique	Range (µM)	LOD (µM)	Interference	Sample	Ref
FLX	AgCFE	CV, CA	0–10	0.079	None	None	[99]
CIT	JUK-2-MWCNTs-AuNPsGCE	CV, EIS	0.05–115	0.011	Organic and inorganic compounds	Tablets, water, urine and serum	[102]
	MIP-hNiNS-AMWCNT@GONRs-GCE	CV, DPV, EIS	0.5–190	0.043		Tablets, urine and serum	[103]
PRX	Poly(ᴅʟ-met)/AuNPs-GCE	CV, DPV, EIS	0.05 nM–100 µM	0.01 nM	Sugars, organic molecules, biological species and amino acids	PRX tablets	[104]
STR	Ni(II)-LD/AuNPs/MWCNTs-GCE	CV	0.05–5.5	0.095	Inorganic and organic molecules, amino acids and sugars	Serum	[64]
	CuMOF/SNDGrPGE	CV, DPV	0.05–2.67	0.038		Blood and tablet	[100]

**Table 8 micromachines-14-01334-t008:** Bio-functionalised polymer-based modification of electrode surface.

Analyte	Coating	Technique	Range (µM)	LOD (µM)	Interference	Sample	Ref
PAR	Poly-2D6	CV	0.005–0.05	0.002	N/A	N/A	[105]

## Data Availability

Submitted herein is a literature review paper. All sources of information used have been referenced.

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
