# Peer review of "Review on Surface-Modified Electrodes for the Enhanced Electrochemical Detection of Selective Serotonin Reuptake Inhibitors (SSRIs)"

_micromachines, 2023, doi:10.3390/mi14071334_

Round 1

Reviewer 1 Report

This review article aims to present functional materials such as polymers, carbon nanomaterials, metal nanomaterials as well as composites for surface modification of electrodes for sensitive detection and quantification of SSRIs, including fluoxetine, citalopram, paroxetine, fluvoxamine and sertraline. The component of this review is comprehensive and interesting, and the outline is suitable. The author has taken a thorough references survey and introduced them well. However, the present edition need some improvement to meet the standard of Micromachines.

In addition to discussing the modification materials, the size and shape of the electrode also need to be discussed, such as SPE or some other microelectrode/sensor. Most of the methods summarized in this paper focus on GCE, its limitations are also obvious, so I think a part of “future trends” or “discussion” part should be added at the end of this paper to discuss the on-line/real-time detection of SSRIs or other applications or such as wearable sensor. The author should compare the respective advantages or shortness of the methods, and raise the questions or challenge of the bio-sensing applications, especially their commercial potentials.

Author Response

Dear Reviewer

The concluding remarks of the review have responded to all the remarks from the reviewer.

Reviewer 2 Report

The authors review several typical functional materials modified electrodes for quantitative detection of SSRIs in various types of samples. These functional materials mainly include polymers, carbon nanomaterials, metal and metal oxide nanomaterials, carbon metal nanocomposites, carbon-polymer composites and carbon-metal-polymer composites.

Major revision is needed.

1. For review, more graphic summary or frame diagram are suggested to be provided.

2.Some self-insights and summary statements can be added at the key points,such ascarbon-metal matrix composites, carbon-polymer matrix composites.

3.Line 573, It should be Table 5 instead of Table 1.

4.Please unify the abbreviations, such as MWCNTs.

5. Please check the super/subscript and upper/lower case.

Author Response

Dear Reviewer

Images have been added to the review article. These can be seen between page 5 and page 19.

The suggested changes have also been included.

Reviewer 3 Report

The manuscript can be pubblished as it is after minor revisions to English 

Author Response

Dear Reviewer

Thank you for your comments.

Reviewer 4 Report

The review paper described by Barry et al. presented functional materials such as polymers, carbon nanomaterials, metal nanomaterials as well as composites for surface modification of electrodes for sensitive detection and quantification of SSRIs, including fluoxetine, citalopram, paroxetine, fluvoxamine and sertraline. The topic is interesting, but the paper is lacking scientific insight and is very crude and elaborately prepared. Thus, the paper is far more than enough that can be accepted.

1.      First of all, it is very rare to see a review paper that did not include any figures. Figure to show the main idea of the paper is needed. And figures of the most used functional materials are also suggested to be provided.

2.      Another serious problem of this review paper is its logic. From section 2 to section 7, the authors seem to summarize the materials used as electrodes. But as a reader, we want to see what kind of materials can be used, and why they are used. However, it’s quite confusing that the following text in each section does not show the materials, but concentrates on the drugs.

3.      Therefore, the review needs extensive rewrites before it can be considered for publication.

Author Response

  1. First of all, it is very rare to see a review paper that did not include any figures. Figure to show the main idea of the paper is needed. And figures of the most used functional materials are also suggested to be provided.

Figures have been added to the manuscript. They are found between pages 5 and 19.

  1. Another serious problem of this review paper is its logic. From section 2 to section 7, the authors seem to summarize the materials used as electrodes. But as a reader, we want to see what kind of materials can be used, and why they are used. However, it’s quite confusing that the following text in each section does not show the materials, but concentrates on the drugs.

More details has been added to the manuscript and more information about the sensing platforms have been added.

  1. Therefore, the review needs extensive rewrites before it can be considered for publication.

We believe that the manuscript has worthy of submission in its current state.

Round 2

Reviewer 2 Report

 Accept in present form

Reviewer 4 Report

The authors have made significant reversions to this review paper, it's now much better and can be accepted after minor reversions.

The quality of the figures is poor, and some of them need to be refined.